# The impact of heating, ventilation, and air conditioning design features on the transmission of viruses, including the 2019 novel coronavirus: A systematic review of ventilation and coronavirus

**Gail M. Thornton**[1], **Brian A. Fleck**[1]*, **Emily Kroeker**[1], **Dhyey Dandnayak**[1], **Natalie Fleck**[1], **Lexuan Zhong**[1], **Lisa Hartling**[2]

1 Faculty of Engineering, Department of Mechanical Engineering, University of Alberta, Edmonton, Canada,
2 Faculty of Medicine & Dentistry, Department of Pediatrics, University of Alberta, Edmonton, Canada

* brian.fleck@ualberta.ca

**Data Availability Statement:** Our paper reports a systematic review of previously published studies.

## Abstract

Aerosol transmission has been a pathway for the spread of many viruses. Similarly, emerging evidence has determined aerosol transmission for Severe Acute Respiratory Syndrome coronavirus 2 (SARS-CoV-2) and the resulting COVID-19 pandemic to be significant. As such, data regarding the effect of Heating, Ventilation, and Air Conditioning (HVAC) features to control and mitigate virus transmission is essential. A systematic review was conducted to identify and comprehensively synthesize research examining the effectiveness of ventilation for mitigating transmission of coronaviruses. A comprehensive search was conducted in Ovid MEDLINE, Compendex, Web of Science Core to January 2021. Study selection, data extraction, and risk of bias assessments were performed by two authors. Evidence tables were developed and results were described narratively. Results from 32 relevant studies showed that: increased ventilation rate was associated with decreased transmission, transmission probability/risk, infection probability/risk, droplet persistence, virus concentration, and increased virus removal and virus particle removal efficiency; increased ventilation rate decreased risk at longer exposure times; some ventilation was better than no ventilation; airflow patterns affected transmission; ventilation feature (e.g., supply/exhaust, fans) placement influenced particle distribution. Few studies provided specific quantitative ventilation parameters suggesting a significant gap in current research. Adapting HVAC ventilation systems to mitigate virus transmission is not a one-solution-fits-all approach. Changing ventilation rate or using mixing ventilation is not always the only way to mitigate and control viruses. Practitioners need to consider occupancy, ventilation feature (supply/exhaust and fans) placement, and exposure time in conjunction with both ventilation rates and airflow patterns. Some recommendations based on quantitative data were made for specific scenarios (e.g., using air change rate of 9 h$^{-1}$ for a hospital ward). Other recommendations included using or increasing ventilation, introducing fresh air, using maximum

All data appears in evidence tables within the paper.

**Funding:** This work is funded by a Canadian Institutes of Health Research (CIHR) Operating Grant: Canadian 2019 Novel Coronavirus (COVID-19) Rapid Research Funding Opportunity [https://webapps.cihr-irsc.gc.ca/decisions/p/project_details.html?applId=422567&lang=en] and Alberta Innovates. The funders had no role in study design, data collection and analysis, decision to publish, or preparation of the manuscript. Author LH is supported by a Canada Research Chair in Knowledge Synthesis and Translation. Authors BF and LZ are supported by the NSERC Discovery program.

**Competing interests:** This work is funded by The Canadian Institutes of Health Research and Alberta Innovates. There are no patents, products in development or marketed products to declare. This does not alter our adherence to PLOS policies on sharing data and materials. The authors have declared that no other competing interests exist.

supply rates, avoiding poorly ventilated spaces, assessing fan placement and potentially increasing ventilation locations, and employing ventilation testing and air balancing checks.

**Trial registration**: PROSPERO 2020 CRD42020193968.

## Introduction

On April 5, 2021, the American Society of Heating, Refrigerating, and Air-Conditioning Engineers (ASHRAE) declared that "Airborne transmission of SARS-CoV-2 is significant and should be controlled. Changes to building operations, including the operation of heating, ventilating, and air-conditioning systems, can reduce airborne exposures" [1]. This statement declaring that airborne transmission of Severe Acute Respiratory Syndrome coronavirus 2 (SARS-CoV-2) is "significant" replaced the April 2020 statement that airborne transmission of SARS-CoV-2 is "sufficiently likely"; however, both statements indicate that heating, ventilation, and air conditioning (HVAC) systems can influence airborne exposures. These ASHRAE statements represent an evolution in the understanding of coronavirus transmission since March 2020 [2] when the World Health Organization declared Coronavirus Disease 2019 (COVID-19), the disease caused by the SARS-CoV-2 [3], a pandemic.

International public health authorities have sought evidence regarding transmission routes and appropriate public health measures to mitigate virus spread since March 2020. Certain viruses are capable of aerosol transmission [4], which can occur when virus-laden aerosols are exhaled by humans and remain airborne for extended periods of time. Recent evidence suggests that SARS-CoV-2 can spread via airborne transmission, particularly in indoor environments with poor ventilation where the air inside is not exchanged with outdoor or fresh air enough to mitigate exposure risks, such as settings with low ventilation rates or areas with high concentrations of viruses or potentially infected air [5,6], further emphasizing the important role of HVAC in virus transmission. Appropriate measures for protecting occupants of indoor spaces based on informed, interdisciplinary research are critical to managing the spread of infectious disease [7].

While seven human coronaviruses have been identified, three have received the most attention because of their pathogenicity and lethality [8]: SARS-CoV-2, Severe Acute Respiratory Syndrome coronavirus (SARS-CoV), and Middle East Respiratory Syndrome coronavirus (MERS-CoV). Each of these coronaviruses had its first emergence in the last 18 years [8]: SARS-CoV in 2003, MERS-CoV in 2012 and SARS-CoV-2 in 2019. Coronavirus has emerged as an infectious agent of great concern for potential airborne transmission.

HVAC systems can reduce airborne virus exposure through dilution or removal of contaminated air inside the building envelope where humans breathe [7,9–11]. Virus transmission can be influenced by various HVAC design features, including ventilation, filtration, ultraviolet radiation, and humidity. Previous systematic reviews that examined HVAC systems and airborne transmission of infectious agents highlighted the need to quantify the HVAC parameters to minimize transmission. Li et al found sufficient evidence to demonstrate an association between transmission of infectious agents and ventilation rate and/or airflow pattern [9]. However, they found insufficient evidence to specify and quantify the minimum ventilation requirements in buildings in relation to the airborne transmission of infectious agents [9]. Similarly, Luongo et al demonstrated an association between increased infectious illness and decreased ventilation rate; however, insufficient data were found to quantify how mechanical ventilation may affect the airborne transmission of infectious agents [7]. Furthermore, a recent review by Shajahan et al reinforced the need to quantify the optimum range for HVAC

parameters considering airborne exposure [12]. At this time, what remains unknown is the specific quantity of any particular HVAC design feature that is effective in reducing virus transmission.

The current systematic review examined whether virus transmission is affected by HVAC design features, particularly, ventilation. In this review, published research evaluating the effectiveness of ventilation in reducing coronavirus transmission was examined. The insight drawn from this review could help answer questions of the utility of ventilation to mitigate the transmission of SARS-CoV-2 in mechanically ventilated indoor environments. Further, understanding effectiveness relative to ventilation rate and airflow patterns could inform control measures.

## Methods

As an integral part of a larger research program to identify and synthesize the scientific literature on airborne virus transmission and HVAC design features, this systematic review focused specifically on the impact of ventilation on coronavirus transmission. Owing to the volume and heterogeneity of the published research, results for the impact of other HVAC design features of interest on virus transmission are reported elsewhere. The *a priori* systematic review protocol is publicly available [13] and the systematic review is registered [14]. The review adheres to the standards for the conduct of systematic reviews defined by the international Cochrane organization [15] with modifications for questions related to etiology [16], and the accepted standards for reporting [17].

### Search strategy

Using concepts for virus, transmission, and HVAC, a research librarian (GMT) searched three electronic databases (Ovid MEDLINE, Compendex, Web of Science Core) from inception to April 2020 with an update in January 2021. Search strategies were peer reviewed by two librarians (TL, AH) prior to implementing the searches. The search strategy for Ovid MEDLINE is provided in Table 1. Screening of reference lists of all relevant papers as well as relevant review articles was undertaken. Conference abstracts, identified through Compendex and Web of Science, were not included but searches were conducted to identify whether any potentially relevant conference abstracts had been published as peer-reviewed journal articles. Year and language limitations were not applied to the search; however, due to the volume of available literature and resource constraints, only English language studies were included. EndNote was utilized to manage records and duplicate records were removed prior to screening.

### Study selection

Study selection was completed in two stages. First, the titles and abstracts of all references identified by the electronic databases searches were screened by two reviewers independently. Each record was classified based on relevance as Yes, Maybe, or No. One reviewer resolved conflicts between Yes/Maybe and No. After each round of pilot testing using three sets of studies (n = 199 each), the review team met to discuss discrepancies and develop decision rules. Second, the full text articles were reviewed and the inclusion and exclusion criteria were applied by two reviewers independently. Each study was classified as Include or Exclude. Consensus of the reviewers resolved conflicts between Include and Exclude. One reviewer resolved conflicts between different exclusion reasons. After each round of pilot testing with three sets of studies (n = 30 each), the review team met to resolve discrepancies. Study selection was conducted using Covidence software.

**Table 1. Search strategy for Ovid MEDLINE ALL 1946 to Present[13].**

| # | Searches |
|---|---|
| 1 | exp Aerosols/ |
| 2 | Air Microbiology/ |
| 3 | exp Viruses/ |
| 4 | (aerosol or aerosols or bioaerosol or bioaerosols).mp. |
| 5 | droplet nuclei.mp. |
| 6 | infectio*.mp. |
| 7 | (pathogen or pathogens).mp. |
| 8 | (virus or viruses or viral or virome).mp. |
| 9 | or/1-8 [MeSH + Keywords–Virus concept] |
| 10 | Air Conditioning/ |
| 11 | Air Filters/ or Filtration/ |
| 12 | Humidity/ |
| 13 | Ventilation/ |
| 14 | Ultraviolet Rays/ |
| 15 | air condition*.mp. |
| 16 | (air change rate or air change rates or air changes per hour or air exchange rate or air exchange rates or air exchanges per hour).mp. |
| 17 | (airflow or air flow).mp. |
| 18 | built environment.mp. |
| 19 | computational fluid dynamics.mp. |
| 20 | ((distance adj6 index) or long distances).mp. |
| 21 | HVAC.mp. |
| 22 | (filter or filters or filtration).mp. |
| 23 | humidity.mp. |
| 24 | (ultraviolet or UV).mp. |
| 25 | ventilat*.mp. |
| 26 | or/10-25 [MeSH + Keywords–HVAC concept] |
| 27 | Air Pollution, Indoor/ |
| 28 | exp Disease Transmission, Infectious/ |
| 29 | (indoor adj1 (air quality or environment*)).mp. |
| 30 | transmission.mp. |
| 31 | or/27-30 [MeSH + Keywords–Transmission concept] |
| 32 | 9 and 26 and 31 |
| 33 | remove duplicates from 32 |

MeSH = Medical Subject Headings.

## Inclusion and exclusion criteria

The inclusion and exclusion criteria are provided in Table 2. Given that this systematic review was part of a larger research program to examine virus transmission and different HVAC design features, searching and screening for all HVAC design features was conducted at once; however, only studies evaluating ventilation were synthesized in this paper. Likewise, searching and screening considered a variety of agents (e.g., bacteria, fungi) but prioritized studies of viruses or agents that simulated viruses. For this specific review, the synthesis was further narrowed from viruses to coronaviruses. Studies of the indoor built environment (e.g., office, public, residential buildings) with mechanical ventilation were of interest. Primary research with quantitative results of correlation or association between ventilation and coronavirus

**Table 2. Inclusion and exclusion criteria for systematic review [13].**

| Item | Inclusion criteria | Exclusion criteria |
|---|---|---|
| Agent | • Viruses<br>• Aerosols<br>• Bioaerosols<br>• Droplet nuclei<br>• Other pathogens (e.g., bacteria, fungi)<br>*We planned a staged process: if we identified studies specific to viruses for each HVAC design feature, we would not include other pathogens; however, for design features where we did not find studies specific to viruses, we would expand to other pathogens.* | |
| HVAC | Design features relating to:<br>• Ventilation (ventilation rate, air changes per hour (ACH), air exchange, airflow pattern, pressurization)<br>• Filtration (air filtration, filter type, MERV rating, filter age and/or use, pressure drop, holding capacity, replacement, change frequency)<br>• Ultraviolet germicidal irradiation (UVGI; power, dose, uniformity of dose, flow rate, bioaerosol inactivation efficiency, location)<br>• Humidity or relative humidity | Examines HVAC / mechanical / or other ventilation mechanisms overall, but not by specific design features. |
| Setting | • Office buildings<br>• Public buildings (e.g., schools, day cares)<br>• Residential buildings<br>• Hospitals and other healthcare facilities (e.g., clinics)<br>• Transport vehicles (e.g., aircraft) or hubs (e.g., airports) | • Outdoor settings<br>• Indoor settings with natural ventilation |
| Outcomes | Quantitative data evaluating the correlation or association between virus transmission and above HVAC features | Qualitative data |
| Study design | Primary research, including:<br>• Epidemiological studies<br>• Observational studies (e.g., cohort, case-control, cross-sectional)<br>• Experimental studies (including human or animal)<br>• Modelling studies, including CFD | • Review articles<br>• Commentaries, opinion pieces<br>• Qualitative studies |
| Language | English<br>*We planned a staged process where we would include studies in languages other than English if we do not identify English language studies for specific HVAC design features or if we identified clusters of potentially relevant studies in another language.* | |
| Year | No restrictions | |
| Publication status | Published, peer-reviewed | Unpublished, not peer-reviewed |

ACH = air changes per hour; CFD = computational fluid dynamics; HVAC = heating, ventilation, and air conditioning; MERV = minimum efficiency reporting value; UVGI = ultraviolet germicidal irradiation.

transmission was included. English-language, peer-reviewed publications were included with no limitation on year of publication.

### Risk of bias assessment

Different risk of bias assessments tools were used for experimental and modelling studies. Risk of bias for experimental studies considered three domains: selection bias, information bias and confounding [18,19]. Each domain was assessed as low, high, or unclear risk of bias using signaling questions [20] from guidance documents for the different study types, e.g., animal studies, laboratory experiments, epidemiological studies [18,19,21]. Risk of bias for modelling studies considered three domains: definition, assumption, and validation [21,22]. Definition evaluated model complexity and data sources, assumption evaluated the description and explanation of model assumptions, and validation evaluated model validation and sensitivity analysis [22]. Each domain was assessed as low, high, or unclear risk of bias using signaling questions [21–23]. After pilot testing among three review authors, the risk of bias was assessed

by three reviewers (EK, DD, NF) independently and discrepancies were resolved by the review team.

## Data extraction

General information was extracted regarding the study (authors, year of publication, country of corresponding author, study design) and methods (setting, population [as applicable], agent studied, intervention set-up). Details on ventilation parameters were extracted, including ventilation rate and airflow pattern. Ventilation rate may be expressed as air changes per hour (ACH), ventilation flow rates ($m^3$/h, $m^3$/min, L/min), ventilation usage (ventilation versus no ventilation), or as determined by carbon dioxide ($CO_2$) levels (ppm). Quantitative data were extracted as well as results of relevant tests of statistical significance related to ventilation. The primary outcome of interest was quantitative measures of the association between ventilation and coronavirus transmission. Data were extracted on actual coronavirus transmission where available (i.e., infections), as well as virus removal, virus concentrations, particle dispersion, and particle persistence, probability of transmission and transmission risk (referred to as transmission probability/risk) and infection risk, infection transmission probability, infection probability, probability of infection, individual risk, and infection index (referred to as infection probability/risk) as applicable. Information regarding ventilation feature placement, supply/exhaust ratios, occupancy, filtration usage (as provided), and air balancing was also extracted. Employing a data extraction form spreadsheet to ensure comprehensiveness and consistency, one reviewer extracted data and a second reviewer verified data for accuracy and completeness. Discrepancies were discussed by the review team.

## Data synthesis

Due to heterogeneity across studies in terms of study design, ventilation features examined, outcomes assessed, and results reported, meta-analysis was not possible as anticipated. Evidence tables describing the studies and their results were developed. Narrative synthesis of the results of relevant studies was provided alongside evidence tables describing the studies and their results. To facilitate meaningful synthesis and comparison across studies, studies were separated into three groups: ventilation use, airflow pattern, and ventilation rate and airflow pattern.

# Results

12,177 unique citations were screened, where 2,428 were identified as potentially relevant from the title and abstract screening and 568 met the inclusion criteria (Fig 1). Of the 568, 332 were relevant to ventilation. Of the 332 relevant to ventilation, 217 were relevant to viruses and, of those, 32 were relevant to coronaviruses. Two of these relevant studies were related and are considered as one in the following syntheses [24,25]. Attempts were made to divide studies into tables examining ventilation rate or airflow pattern. Most studies examining both ventilation rate and airflow pattern were challenging to separate into either individual category and a third table was created: combined effect of ventilation rate and airflow pattern. However, the study by Shao et al could be separated into individual categories and appears in the ventilation rate table and the airflow pattern table [26]. Thus, 20 investigated ventilation rate, eight investigated airflow patterns, and five investigated the combined effect of ventilation rate and airflow patterns. Studies were published between 2004 and 2021 (median year 2020). While the SARS-CoV-2 studies were concentrated between 2020 and 2021, the MERS studies were in 2017 and 2020, and the SARS-CoV studies ranged from 2004 to 2020. Most studies were conducted in the United States (n = 11) and China (n = 10). Studies were funded by national

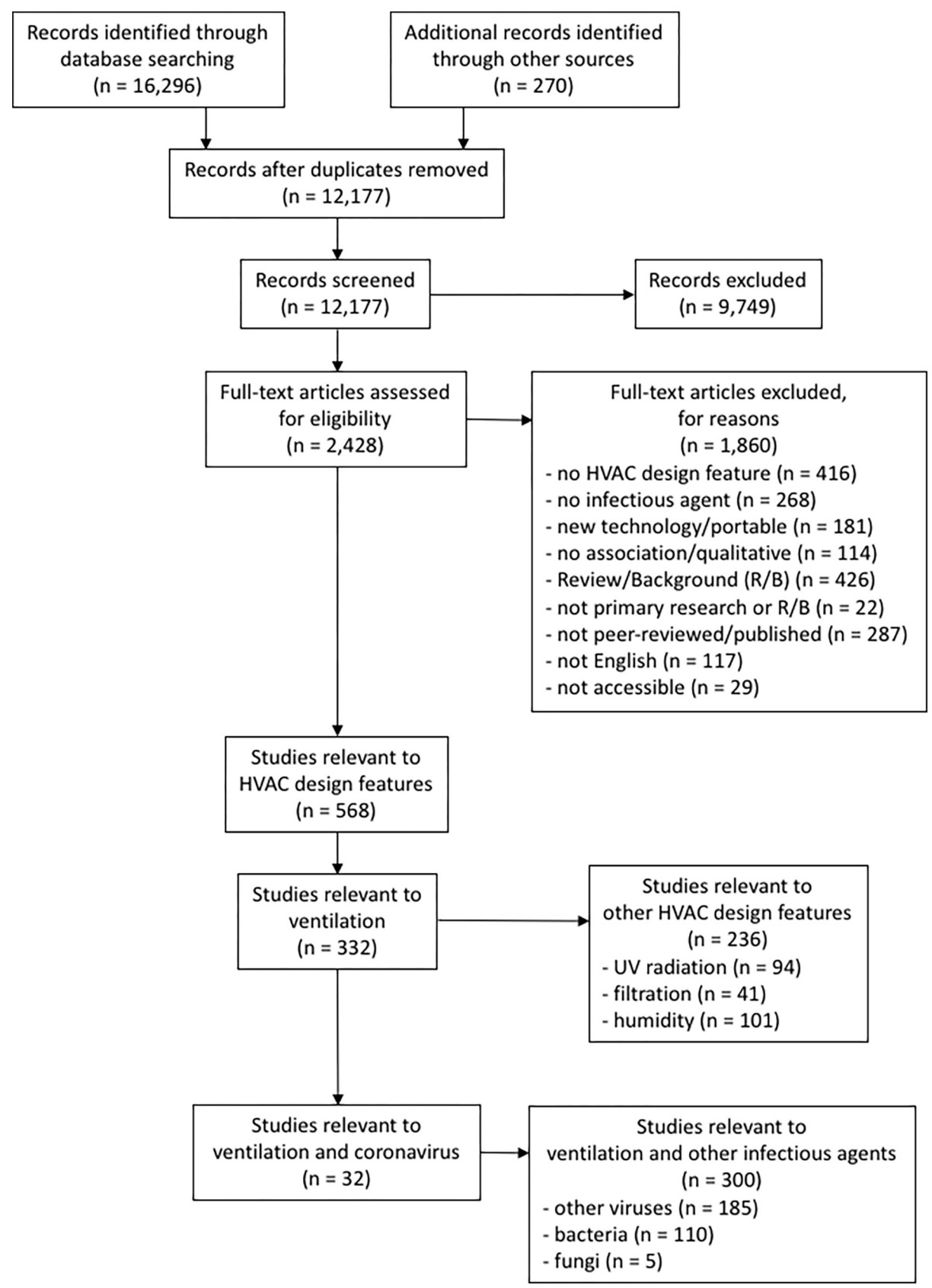

**Fig 1. Flow of studies through the selection process (note: Search was conducted for all HVAC design features but only studies of ventilation and coronavirus are included in this manuscript).**

research funding organizations (n = 15), university and state grants (n = 9), industries (n = 2), and public foundations (n = 2); nine studies did not report funding sources and two studies reported no external funding.

## Ventilation rate

Twenty studies, including modelling (n = 16), experimental (n = 3), and observational (n = 1) studies, analyzed the effect of ventilation rate on SARS-CoV-2/COVID-19 (n = 16),SARS-CoV/SARS (n = 3), and MERS-CoV (n = 2) (Table 3). Scenarios in the studies represented a variety of settings, including hospitals [27–33], schools [27,34], dental clinics [35], office spaces [31,34,36–38], laboratories [36], transport vehicles and hubs [34,38–40], spaces used for singing [41], nail salons [42], conference and meeting rooms [32,38], experimental test set-ups [43], gyms [32,38], restaurants and nightclubs [32,38], elevators [26,38], and rooms, in both single [33,37,38,44,45] and multi-room facilities [45].

Of the 16 modelling studies analyzing ventilation rate, six studies found that increased ventilation (i.e., increased ACH) was associated with decreased transmission [27], virus concentration [31], probability of infection [36], infection risk [29,37], and risk of cross infection [44]. Two modelling studies found that increased ACH increased the efficiency of particle and virus removal [26,28]. Shao et al also found that increasing ventilation by using increased ventilation settings (i.e., more ventilation sites) was effective in particle removal [26]. Additionally, four modelling studies found that increasing ventilation rate ($m^3/h$ and $m^3/min$) was associated with decreased infection probability [34,39], risk of airborne transmission [42], and

**Table 3. Characteristics and results of studies examining ventilation rate and coronavirus transmission.**

| Author Year Country | Outcome parameter | Data | Association |
|---|---|---|---|
| Liao (2005) [27] Taiwan Modelling; Hospital, elementary school SARS | Ventilation (indicated by $CO_2$ ppm and ACH) vs transmission (based on $R_0$) | Reproductive Number ($R_0$) for 6-hour exposure time (from Fig 6): <br>~5.4 at 0.02 ACH <br>~4.7 at 2.43 ACH <br>~4.0 at 4.84 ACH <br>~3.1 at 7.25 ACH <br>~2.3 at 9.66 ACH <br>~1.4 at 12.07 ACH <br>~0.5 at 14.48 ACH <br>Reproductive Number ($R_0$) for 1-hour exposure time (from Fig 6): <br><1 for 0.02–14.48 ACH | Increased ventilation (ACH and outdoor air ventilation) associated with decreased transmission |
| Yu (2017) [28] China Modelling; Hospital ward MERS-CoV, SARS-CoV | Ventilation rate (ACH) vs MERS-CoV and SARS-CoV removal <br>$r_e$ = exhausted ratio <br>$\tau_a$ = elapsed time | Elapsed time required for all virus particles to be exhausted or deposited <br>$4h^{-1}$: 70 s <br>$6h^{-1}$: 61 s <br>$9h^{-1}$: 48 s <br>$12h^{-1}$: 34 s <br>"higher $r_e$ and shorter $\tau_a$ were found with increasing air change rates" (p.522) | Increased ventilation rate (ACH) associated with decreased elapsed time $\tau_a$ and increased virus removal <br>Elapsed time associated with inhalation risk (decreased elapsed time means decreased inhalation risk) <br>*Authors' Recommendations*: <br>"For a typical semi-enclosed six-bed general ward of Hong Kong hospitals, an air change rate of 9 $h^{-1}$ could effectively minimize the deposition and floating time of respiratory virus particles while maximizing energy efficiency." (p.526) |

*(Continued)*

**Table 3.** (Continued)

| Author Year Country | Outcome parameter | Data | Association |
|---|---|---|---|
| Adhikari (Sept 2019) [29] USA Modelling; hospital MERV-CoV | Ventilation rate (ACH) vs. risk of infection | Median (mean) daily risk of infection at 6 ACH Index patient and other patients in the room: $1.33 \times 10^{-8}$ ($8.49 \times 10^{-4}$) HCW: $1.18 \times 10^{-8}$ ($7.91 \times 10^{-4}$) Visiting family members: $6.36 \times 10^{-9}$ ($3.12 \times 10^{-4}$) Other patients sharing the room: $2.73 \times 10^{-9}$ ($1.29 \times 10^{-4}$) -"the daily risk of infection for healthcare workers was *significantly higher* than the one for the other patients or the family visitors" (p.2617) -"comparing nurses and other healthcare worker, the result is not significant" (p.2617) -"Other patients in the same room had a statistically *significant lower* risk of infection compared to nurses. . . but had *nonsignificant* statistical differences in risk with family visitors" (p,2617) Comparison of risk reduction per intervention (based on 1-day 6 ACH base scenario) [ACH: % of average daily risk of infection] Nurses: -6 ACH: 100% -9 ACH: 98.10% -12 ACH: 97.48% HCW: -6 ACH: 100% -9 ACH: 98.42% -12 ACH: 97.69% Family: -6 ACH: 100% -9 ACH: 98.34% -12 ACH: 98.22% Other patients: -6 ACH: 100% -9 ACH: 70.66% -12 ACH: 52.87% -"For the other patients, mean daily risk of infection could be reduced by about 30% or 58% through increasing the air ventilation from 6 to 9 or 12 ACH" (p.2619) -"For the nurses, healthcare workers, and family visitors, only up to about 2% reduction in mean daily risk could be achieved by increasing the ACH from 6 to 12." (p.2619) | Increased ventilation rate (ACH) associated with decreased risk of infection |
| Somsen (May 2020) [30] Netherlands Experimental; hospital SARS-CoV-2 | Ventilation use vs droplet persistence/ airborne time of respiratory droplets $t_{1/2}$ = time to halve number of droplets in the air | No ventilation: $t_{1/2} \approx 5.3$ min Mechanical (poor) ventilation: $t_{1/2} \approx 1.4$ min | Increased ventilation (ventilation vs no ventilation) associated with decreased airborne respiratory droplet persistence *Authors' Recommendations*: "health-care authorities should consider the recommendation to avoid poorly ventilated public spaces as much as possible." (p.659) |

(*Continued*)

**Table 3.** (Continued)

| Author Year Country | Outcome parameter | Data | Association |
|---|---|---|---|
| Zemouri (Jul 2020) [35] Netherlands Modelling; Dental clinic SARS-CoV | Level of ventilation (determined by $CO_2$ levels) vs Infection transmission probability Low risk: 774 ppm Intermediate risk: 1135 ppm High Risk: 2375 ppm -Level of $CO_2$ indicative of ventilation levels; improved ventilation results in decreased $CO_2$ | Infection transmission probability, % Low risk: 0.0% Intermediate risk: 13.1% High risk: 99.44% | Increased ventilation (lower $CO_2$ levels) associated with decreased infection transmission probability |
| Riediker (Jul 2020) [31] Switzerland Modelling; Hospital, office SARS-CoV-2 | Ventilation rates (ACH) vs Concentration and concentration plateau (concentration over time) | Time until 99% plateau, minutes 1 ACH: 169 minutes 3 ACH: 77 minutes 10 ACH: 26 minutes 20 ACH: 14 minutes Airborne viral concentrations at plateau, copies/m³ (regular breathing: low; typical; high emitter) 1 ACH: 0.000009598; 0.009598; 1247.7 3 ACH: 0.000004310; 0.004310; 560.3 10 ACH: 0.000001472; 0.001472; 191.3 20 ACH: 0.000000758; 0.000758; 98.6 (frequent coughing: low; typical; high emitter) 1 ACH: 0.057251; 57.251; 7 442 598 3 ACH: 0.025709; 25.709; 3 342 148 10 ACH: 0.008779; 8.779; 1 141 326 20 ACH: 0.004524; 4.524; 588 093 | Increased ventilation rate (ACH) associated with decreased virus concentration plateau time Increased ventilation rate (ACH) associated with decreased airborne viral concentration at plateau for regular breathing and frequent coughing |
| Dai (Aug 2020) [34] China Modelling; Office, classroom, bus, and aircraft cabin COVID-19 | Ventilation rate (m³/h) vs Infection probability vs exposure time | -Ventilation rate of 100–350 m³/h per infector was required to ensure an infection probability of less than 1% for 0.25 h of exposure of a susceptible person -Ventilation rate of 1200–4000 m³/h per infector was required to ensure an infection probability of less than 1% for 3 h of exposure of a susceptible person Examples from Table 2 (data also available for bus and aircraft cabin scenarios): Classroom (348 m³) with 2 h exposure time (infection probability: ACH) $q = 14$ h⁻¹ 0.1%: 20.00 ACH 1.0%: **2.40 ACH** 2.0%: **1.15 ACH** $q = 48$ h⁻¹ 0.1%: 71.00 ACH 1.0%: **7.00 ACH** 2.0%: **4.00 ACH** Office (150 m³) with 8 h exposure time (infection probability: ACH) $q = 14$ h⁻¹ 0.1%: 200.00 ACH 1.0%: 20.00 ACH 2.0%: 10.00 ACH $q = 48$ h⁻¹ 0.1%: 666.00 ACH 1.0%: 80.00 ACH 2.0%: 36.00 ACH *ACH in bold were identified by the authors as ventilation rates that could be achieved by typical ventilation systems | Increased ventilation rate associated with decreased infection probability For longer exposure time, increased ventilation rate associated with decreased infection probability |

(*Continued*)

**Table 3.** (Continued)

| Author Year Country | Outcome parameter | Data | Association |
|---|---|---|---|
| Augenbraun (Sept 2020) [36] USA Modelling; laboratory, office SARS-CoV-2 | Ventilation rate (ACH/FCH) vs Probability of transmission | Office ~2 FCH 99% probability of infection for 1 week of exposure. Laboratory ~12 ACH with HEPA <1% probability of infection for 3 weeks of total exposure in a 6-month period of work | Increased ventilation rate (ACH/FCH) prior to occupancy of room associated with decreased probability of infection *Authors' Recommendations:* "We find that for environments with HVAC systems typical of laboratories and offices, it is safe to operate when a room (or section of a room with an isolated airstream) is left vacant for one (high-circulation HVAC with HEPA filtration) to six (low-circulation with no filtration) air exchange times before a new worker enters." (p.453) Office: "For a typical office room with 2 fresh air changes per hour, this will be approximately 2.5 hr (5 hr) for 3 weeks (26 weeks) of total exposure over 6 months." (p.453) Lab: "For a typical HEPA filtered lab this would be one air change (for either 3 or 26 weeks of exposure in a 6-month period). For shared lab resources (e.g., electronics rooms, storage cabinets, chemical rooms, etc.) without HEPA filtration, a wait time of at least four air changes will be required ([supplementary material] Sec. A.5), for long exposure times." (p.453) |
| Miller (Sept 2020) [41] USA Modelling; public buildings (spaces used for singing) COVID-19/SARS-CoV-2 | Viral aerosol loss rates (dependent on ventilation rate, metric unspecified) vs Probability of infection | For the mean value emission rate = 970 quanta/h, increasing the loss rate coefficient from a nominal baseline value of 0.6 to $5h^{-1}$ would reduce the probability of infection by a factor greater than two, from 91% to 42%. For the full range of loss rates plotted in Fig 1, the infection risks span a factor of eight: from 98% to 13%. For durations ranging from 0.5 to 2.5 hours, and total loss rates ranging from 0.6 to $12h^{-1}$, the predicted percentage infected spanned a broad extent, from 4% to 91%. At an emission rate of 960 quanta/h probability of infection is reduced from 91% to 42% when loss rate (a factor including deposition, ventilation and inactivation) is increased (Fig 1) | Increased ventilation (loss rate) associated with decreased probability of infection; loss rate due to ventilation increase from $0.3–1.0h^{-1}$ |
| Harrichandra (Oct 2020) [42] USA Modelling; nail salons SARS-CoV-2 | Ventilation rate (m³/min) vs transmission risk Scenario 3: One susceptible customer and one infected customer enter the nail salon together and both stay for 30 min. Scenario 4: One infected customer enters and stays for 45 min, while one susceptible customer enters 30 min after the infected customer and stays for 60 min. Scenario 5: One infected customer and one susceptible customer enter at the same time and both stay for 150 min (2.5 h). | Risk of airborne infection transmission (%) [Salon # (outdoor flow rate): scenario 3; scenario 4; scenario 5] 1 (14.1): 4.27%; 0.68%; 7.690% 2 (5.17): 9.71%; 3.26%; 19.58% 3 (3.72): 8.84%; 9.83%; 25.47% 4 (6.06): 7.43%; 3.83%; 16.91% 5 (5.9): 7.43%; 4.18%; 17.31% 6 (9.46): 3.17%; 4.42%; 10.71% 7 (10.24): 7.69%; 0.10%; 10.43% 8 (21.99): 2.59%; 0.70%; 5.0000% 9 (11.89): 4.02%; 1.78%; 9.020% 10 (6.99): 5.79%; 4.17%; 14.75% 11 (9.8): 4.36%; 2.75%; 10.79% -During an 8-hour workday the average airborne infection transmission risk was reduced from 99.25% to 56.24% when outdoor airflow rate is increased from 3.72m³/min to 21.99m³/min -During 60-minute exposure the average infection transmission risk was reduced from 45.71% to 9.82% when outdoor airflow rate is increased from 3.75m³/min to 21.99m³/min | Increased ventilation rate (outdoor flow rate) associated with decreased risk of airborne transmission Increased ventilation associated with decreased risk for different exposure times |

*(Continued)*

**Table 3.** (Continued)

| Author Year Country | Outcome parameter | Data | Association |
|---|---|---|---|
| Zhang (Oct 2020) [37] Switzerland Modelling; office SARS-CoV-2 | Ventilation rate (ACH) vs Infection risk | Infection risk 0.1 ACH: $5.2 \times 10^{-5}$ 1 ACH: $2.3 \times 10^{-5}$ 2 ACH: $< 2.3 \times 10^{-5}$ -Increasing ACH from 1 ACH to 9 ACH made the risk about 3 times lower | Increased ventilation risk (ACH) associated with decreased infection risk |
| Sun (Nov 2020) [39] China Modelling; bus, airplane COVID-19 | Ventilation rate vs infection probability; occupancy vs minimum ventilation requirements | Location; exposure time; ventilation rate; probability of infection Subway; 0.5h; 20 $m^3$/h·person; 11.3% Classroom; 0.75h; 14 $m^3$/h·person; 15.1% Public Bus; 0.5h; 15 $m^3$/h·person; 17.0% Restaurant; 1.5h; 20 $m^3$/h·person; 24.9% Office; 4h; 30 $m^3$/h·person; 25.6% Air cabin; 2.5h; 25 $m^3$/h·person; 29.3% Long bus; 2h; 20 $m^3$/h·person; 29.9% High-speed train; 3h; 20 $m^3$/h·person; 37.5% | Increased ventilation rate was associated with decreased infection probability. Reduced occupancy associated with decreased minimum ventilation requirements |
| Melikov (Nov 2020) [44] China Modelling; room COVID-19 | Ventilation (ACH) vs Time average intake fraction (risk of cross infection) | Change in intake fraction: 0 ACH: +546% 0.9 ACH: +174% 3.4 ACH: 0% (reference case) 7.1 ACH: -45% 14.5 ACH: -69% 21.9 ACH: -79% | Increased ventilation rate (ACH) reduces the intake fraction (risk of cross infection) *Authors' Recommendations*: "Ventilation systems supplying clean outdoor air should be operated continuously with the maximum supply airflow rate. It is recommended that steady-state conditions in the room in terms of the supply airflow rate should exist at the time when occupation begins, i.e. the ventilation should be in operation before any occupants enter the room" (p.6) |
| Qiao (Dec 2020) [43] USA Experimental; Experimental test set-up SARS-CoV-2 | Ventilation rate (L/min) vs log reduction in viable coronavirus with removal efficiencies | Log reduction of virus titer by UV inactivation: 684 L/min: ~3.73 (99.98% efficient) 1674 L/min: ~3.39 (99.96% efficient) 2439 L/min: ~2.20 (99.40% efficient) | Increased ventilation rate (L $min^{-1}$) associated with lower log reduction and removal efficiencies of viable virus in combination with UV |
| Buonanno (Dec 2020) [32] Italy and Australia Modelling; Hospital room, gym, public buildings (e.g., restaurant), conference room SARS-CoV-2 | Ventilation rate (ACH) v. Individual risk (R) | Time taken to reach acceptable risk reference value of 10-3: Hospital room (exposed subject-medical staff): 39 min (3 ACH); 72 min (10 ACH) Hospital room (exposed subject-patient): 84 min (3 ACH); 192 min (10 ACH) Gym: 13 min (3 ACH); 17 min (10 ACH) Public building (ex. Restaurant): 15 min (3 ACH); 20 min (10 ACH) Conference room: 16 min (3 ACH); 21 min (10 ACH) | Increased ventilation rate (ACH) associated with increasing exposure before reaching accepted risk (AER: $3h^{-1}$ and $10h^{-1}$) |
| Somsen (Dec 2020) [38] Netherlands Experimental SARS-CoV-2 | Ventilation rate (ACH) and droplet persistence times vs transmission risk | Exposure to $<10^3$ microdroplets = low risk, $10^3–10^5$ microdroplets = intermediate risk, $>10^5$ microdroplets = high risk -Restroom (~1 ACH): intermediate -Unventilated living room (~1 ACH): intermediate -Elevator (~1–4 ACH): intermediate -Train (0–5 ACH): low -Gym (5–15 ACH): low -Night club (5–15 ACH): low -Airport (5–15 ACH): low -Car (5–20 ACH): low -Restaurant (8 ACH): low -Meeting room (10 ACH): low -Office space (10 ACH): low | Increased ventilation rate (above ~1–4 ACH) associated with low transmission risk |

*(Continued)*

**Table 3.** (Continued)

| Author Year Country | Outcome parameter | Data | Association |
|---|---|---|---|
| Shao (Jan 2021) [26] USA (see also Table 4) Modelling; Elevator COVID-19 | ventilation rate (ACH) and ventilation location vs risk of particle encounters/ efficiency of particle removal | Elevator 30 ACH: infection risk is extremely low in most of the space 2 ACH: little risk to the people who are not standing near the emitter but two orders of magnitude higher risks for some local hot spots | Increased ventilation rate (ACH) and increased ventilation settings "(e.g., adding more sites of ventilation)" (p.7) associated with increased efficiency of particle removal |
| Moreno (Feb 2021) [40] Spain Observational; Subway and Bus scenarios SARS-CoV-2 | Ventilation rate (usage) vs Probability of infection -Scenario 1: 2 groups of 50 people; 0.2/h AER, recirculated air -Scenario 2: 2 groups of 50 people; 12.6/h AER -Scenario 3: 6 groups of 20 people; 0.2/h AER, recirculated air -Scenario 4: 6 groups of 20 people; 12.6/h AER -Scenario 5: 2 groups of 65 people; 0.2/h AER, recirculated air -Scenario 6: 2 groups of 65 people; 8.4/h AER | Individual infection risk (speaking; breathing) Scenario 1: 0.72%; 0.15% Scenario 2: 0.09%; 0.02% Scenario 3: 0.09%; 0.02% Scenario 4: 0.01%; 0.00% Scenario 5: 0.48%; 0.10% Scenario 6: 0.08%; 0.02% | Increased ventilation rate (ventilation versus no ventilation) associated with decreased individual infection risk *Authors' Recommendations:* "we recommend close attention to be paid to ventilation systems, increasing the forced ventilation rate and the introduction of external air wherever possible (evaluating the result by using $CO_2$ sensors) and improving filtration systems." (p.10) |
| Borro (Feb 2021) [33] Italy Modelling; room, hospital SARS-CoV-2 | Ventilation rate ($m^3$/h) vs infection index η | During 60 secs in Case A: No HVAC, coughed infectious agents only reaches one subject Case B: Nominal flow rate (2020 $m^3$/h from each diffuser) resulted in less concentration affecting the nearest subject Case C: Double flow rate (4040 $m^3$/h from each diffuser) lowered droplet concentration of -99.6% relative to scenario A; lowered droplet concentration up to -77% relative to scenario B, but also resulted in greater dispersion of the infectious agent throughout the space | Increased ventilation rate associated with less infection index to a particular subject in the room Increased ventilation associated with increased turbulent transport and enhanced dispersion of infectious agent throughout the room |
| Kennedy (Mar 2021) [45] USA Modelling; single and multi-region zones SARS-CoV-2 | Ventilation use vs Infection risk | Single region infection risk No ventilation: 7.1% Ventilation: 3.1% Multi region infection risk (region 1; region 2; region 3) No ventilation: 26.4%, <0.1%, 3.6% Ventilation: 12.9%, 4.1%, 0.5% | Increased ventilation rate (ventilation versus no ventilation) associated with decreased infection risk |

ACH = air changes per hour; AER = air exchange rate; $CO_2$ = carbon dioxide; FCH = Fresh air changes per hour; HCW = Healthcare worker; HEPA = High efficiency particulate air; HVAC = Heating ventilation, and air-conditioning; SARS = severe acute respiratory syndrome; SARS-CoV-2: Severe Acute Respiratory Syndrome coronavirus 2; UV = ultraviolet radiation; vs = versus.

infection index η [33]. Sun et al also found that reduced occupancy was associated with lower minimum ventilation requirements [39]. Kennedy et al, when comparing no ventilation scenarios to ventilation scenarios, found that increased ventilation rate, through the use of ventilation systems, was associated with decreased infection risk [45]. Similarly, Miller et al, although not using a specified metric, determined that increased ventilation rate, which led to a subsequent decrease in viral aerosol loss rates, was associated with decreased probability of infection [41].

Two modelling studies used $CO_2$ levels as an indicator of ventilation rate. Both studies found that increased ventilation rate was associated with decreased $CO_2$ levels and, as a result, decreased infection transmission probability [35] and transmission [27]. Two other modelling

studies explored the impact of ventilation rate on individual risk and exposure times [32,34]. Both found that increasing ventilation rate (measured as ACH or $m^3$/h) was associated with longer exposure times before the individual risk exceeded acceptable risk levels [32] or the infection probability increased to 1% [34].

Of the 20 ventilation rate studies, four were experimental or observational studies. Like the modelling study by Kennedy et al [45], two studies found that increasing ventilation rate through the usage of ventilation (ventilation versus no ventilation) was associated with decreased airborne respiratory droplet persistence [30] and decreased individual infection risk [40]. Similar to other modelling studies using ACH [26–29,31,32,36,37,44], Somsen et al found that increased ventilation rates above ~1–4 ACH were associated with decreased transmission risk [38]. One study examined the impact of ventilation rate with UV radiation: Qiao et al found that increasing ventilation rate (flow rate) was associated with lower log reduction and removal efficiencies of viable virus in combination with UV radiation [43]. A more detailed description of the role and influence of UV in virus mitigation was explored in a complementary UV radiation and virus transmission systematic review, resulting from this investigation into HVAC features and the impact on virus transmission [13].

While five ventilation rate studies provided recommendations, only two studies provided quantitative recommendations [28,36]. Yu et al suggested that an air change rate of 9 $h^{-1}$, in a six-bed hospital ward, could potentially decrease elapsed particle removal time, resulting in decreased inhalation risk, while maximizing energy efficiency [28]. Augenbraun et al suggested ACH rates and associated wait times for individuals prior to work or equipment usage in office and laboratory settings [28]. In a typical 200 $ft^2$ (18.58 $m^2$) office room and a volume of 80,000 liters without filtration and ~2 fresh air changes (FCH), they recommended individuals wait six air changes or approximately 2.5 hours before reoccupying the office space [28]. Similar recommendations were made for a laboratory room without filtration: in a 500 $ft^2$ (46.45 $m^2$) laboratory room with a volume of 200,000 liters and High-Efficiency Particulate Absorbing (HEPA) or highly rated Minimum Efficiency Reporting Values (MERV) filtration, they recommended waiting one air change before reoccupying the laboratory room with different users [28]. Recommendations without quantitative metrics included avoiding poorly ventilated spaces [30], ensuring ventilation is operational before occupants enter rooms [44]. Other recommendations suggest increasing ventilation [40], introducing fresh air [40], and using maximum supply rates [44].

## Airflow pattern

Seven studies, including modelling (n = 4), epidemiological (n = 2), and experimental and modelling (n = 1) studies, examined the impact of airflow pattern on SARS-CoV-2/COVID-19 (n = 6) and SARS-CoV/SARS (n = 1) (Table 4). Settings included airplane cabins [46], restaurants [24,25,47], hospitals and healthcare facilities [48–50], supermarkets [26], and classrooms [26].

Two of the four modelling studies determined that the placement of ventilation grilles was important in the dispersion of infectious particles [26,48]. Shao et al showed that when the supply and exhaust grilles were near the infectious particle emitter, the infectious particles were less dispersed [26]. You et al found the infection risk in an airliner cabin to be lower when displacement ventilation was used compared to mixing ventilation [46]. Miller et al determined that viral particles shed in patient rooms of an experimental negative pressure isolation space could spread from room to room or leak into the hallway when the patient room doors are open [49]. Further, viral particles shed in the hallway could infiltrate into patient rooms.

**Table 4. Characteristics and results of studies examining airflow pattern and coronavirus transmission.**

| Author Year; Country | Outcome parameter | Data | Association |
|---|---|---|---|
| You (Sept 2019) [46] China Modelling; airplane cabin SARS | Ventilation type vs average SARS risk of infection | Average SARS infection risk levels among all passengers was 0.23 for mixing ventilation and 0.09 for displacement ventilation. | Mixing ventilation was associated with a higher risk of infection compared to displacement ventilation |
| Anghel (Sept 2020) [48] Romania Modelling; cardiac intensive care unit SARS-CoV-2 | HVAC inlet position vs infectious particle distribution | HVAC inlet in the middle of the ICU. Infected patient on the right side of the ICU: fast dispersion of infecting particles Infected patient on the left side of the ICU: fast dispersion of infecting particles, more dispersed than other two scenarios Infected patient in the middle of the ICU: particles initially carried on the wall and the window behind the bed, then down under the bed, and after that to the office from the center of the ICU. | HVAC inlet position associated with infectious particle distribution *Authors' Recommendations*: "[our study] can provide important recommendations for disease control and optimization of ventilation in intensive care units, by increasing the rate of air change, decreasing recirculation of air and increasing the use of outdoor air and HEPA filters." (p.14) |
| Lu (July, Nov 2020) [24,25] China Epidemiological; Restaurant AC units COVID-19 | Airflow pattern vs transmission of COVID-19 | A total of 10/83 customers infected with COVID-19 10/21 customers at 3 tables in the path of AC airflow tested positive for COVID-19 0/62 customers at 12 tables not in path of AC airflow tested positive for COVID-19 | Airflow pattern associated with transmission of COVID-19 *Authors' Recommendations*: "we recommend. . . increasing the distance between tables and improving ventilation." (Lu (July, 2020, p.1628) |
| Miller (Oct 2020) [49] USA Modelling and experimental; negative pressure isolation space (hospital) SARS-CoV-2 | Negative pressure isolation space vs SARS-CoV-2 transmission | Experiment: As of June 23, 2020, 21 individuals with a confirmed case of COVID-19 were treated in the isolation space which had an average pressure differential of -2.3 Pa between it and the external hallway connected to the rest of the facility. No facility-acquired transmission was identified during this study. Modelling: When patient room doors were open, viral particles may spread from room to room, and into the hallway. Viral particles released in the hallway may also spread into the patient rooms. | A makeshift negative pressure isolation space associated with no transmission of SARS-CoV-2 between isolated residents, staff, or other residents. *Authors' Recommendations*: "We recommend [a pressure difference of] anywhere between -2 and -25 Pa as reasonable." (p.8) |
| Kwon (Nov 2020) [47] South Korea Epidemiological; Restaurant AC units COVID-19 | Airflow pattern vs transmission of COVID-19 | 2/13 contacts with infector (Case B) tested positive for COVID-19 Case B to A: droplet transport potentially 6.5m with 5 minutes of exposure Case B to C: droplet transport potentially 4.8m with 21 minutes of exposure | Airflow pattern associated with transmission of COVID-19 *Authors' Recommendations*: "it is necessary to assess the seating arrangement and operation and location of fans (including ceiling fans) or air conditioners with wind direction and velocity. It is also necessary to ventilation frequently for management of indoor air or to apply a ventilation system or forced ventilation method if natural ventilation is not possible. Furthermore, the distance between tables at an indoor restaurant or cafeteria should be greater than 1–2 m, or installation of a wind partition should be considered based on airflow." (p.6) |
| Shao (Jan 2021) [26] USA (see also Table 3) Modelling; Classroom, supermarket SARS-CoV-2 | Ventilation location vs particle dispersion and risk of infection | Classroom: Ventilation in back corner, far away from the emitter: ventilation spreads particles to the back half of the classroom, student sitting in a hot spot near the vent could inhale several times more particles than a student near the front, increasing their infection risk Ventilation on the same side, near the emitter: spread of particles mostly confined to the front of the classroom before the students. Infection risk significantly reduced compared to ventilation in the back Supermarket: Ventilation in back corner: particles spread across the entire supermarket; a hot spot is formed in the space between the leftmost shelf and corner near the ventilation Ventilation at entrance: overall spread of particles reduced; several hotspots created including one in front of the cashier increasing risk by ~2 orders of magnitude | Ventilation and exhaust located near the infectious particle emitter associated with decreased spread of particles which decreases risk of infection *Authors' Recommendations*: "our results suggest that optimizing ventilation settings (e.g., adding more sites of ventilation) even under the current ventilation capacity can significantly improve the efficiency of particle removal." (p.7) |

*(Continued)*

**Table 4.** (Continued)

| Author Year; Country | Outcome parameter | Data | Association |
|---|---|---|---|
| Ding (Jan 2021) [50] China Experimental; hospital SARS-CoV-2 | Airflow vs dispersion of airborne SARS-CoV-2 | 1 weakly positive air sample for SARS-CoV-2 out of 5 was obtained from the corridor close to the patient's isolation rooms (a total of 46 air samples were taken) | Airflow leakage from the isolation rooms to the corridor associated with one weakly positive SARS-CoV-2 air sample |

AC or A/C = Air-conditioning; HEPA = High-Efficiency Particulate Air; HVAC = Heating, ventilation, and air conditioning; ICU = Intensive care unit; SARS = severe acute respiratory syndrome.

Miller et al examined the dispersion of viral particles in the experimental negative pressure isolation space and found, following the use of careful protective measures, that, as of June 23, 2020, no facility-acquired transmission of SARS-CoV-2 was identified [49]. Ding et al found that airflow leakage from isolation rooms to the corridor of the fifth floor of a hospital (similar to the modelling results from Miller et al [49]) led to one weakly positive air sample of SARS-CoV-2 out of five samples taken from the corridor (a total of 46 air samples were taken from the hospital) [50]. Two studies that retroactively analyzed separate COVID-19 outbreaks in restaurants found that airflow pattern was an essential factor in the transmission of the virus [24,25,47].

Only one study provided quantified recommendations [49]; four studies provided recommendations without quantification [24–26,47,48]. Miller et al recommended using a pressure difference between -2 and -25 Pa in negative pressure isolation spaces [49]. Other recommendations regarding airflow patterns included: increasing air changes and outdoor air usage while decreasing air recirculation [48]; improving ventilation generally [24,25]; assessing fan placement in relation to restaurant seating and ensuring frequent ventilation or applying ventilation systems [47]; and optimizing ventilation through increased ventilation locations [26].

### Ventilation rate and airflow pattern

Five modelling studies analyzed the effect of both ventilation rate and airflow patterns on SARS-CoV-2/COVID-19 (n = 1), SARS-CoV/SARS (n = 3), and MERS-CoV (n = 1) (Table 5). These modelling studies used settings representing hospital wards [51–54], three of which were the Prince of Wales Hospital in Hong Kong [51–53], and a supermarket [55].

Three studies analyzed the 2003 outbreak of SARS in ward 8A of the Prince of Wales Hospital, Hong Kong [51–53]. Two studies agreed that airflow balancing reduced the concentration and dispersion of virus particles [51,52], although Li et al found that the impact was relatively small [51]. In addition, Li et al found that downward ventilation provided a greater reduction in the dispersion of the virus particles compared to mixing ventilation in the imbalanced airflow case [51]. Lim et al found that air supplied from the room boundary and exhausted in the center of the room provided a greater reduction in the dispersion of the virus particles compared to the inverse case in both balanced and imbalanced airflow cases [52]. In agreement with Li et al who found a small effect of airflow balancing on virus particle concentration and dispersion [51], Chen et al determined that airflow balancing had a relatively small to no effect on the dispersion of virus particles in the hospital ward, which was simulated using multi-zone modelling [53].

Satheesan et al found that increasing the ventilation rate greatly reduced the infection risk for patients situated farther away from the corridor within the ward [54]. However, increasing the ventilation rate also resulted in an increase of the infection risk of corridor users and its connected amenities. Installing exhaust grilles close to each patient reduced infection risk

**Table 5. Characteristics and results of studies examining combined effect of ventilation rate and airflow pattern on coronavirus transmission.**

| Author Year; Country | Outcome parameter | Data | Association |
|---|---|---|---|
| Li (Sept 2004) [51] China Modelling; Prince of Wales hospital ward SARS | Ventilation system type and airflow balancing vs distribution of normalized virus-laden bio-aerosol concentrations | Normalized tracer gas ($CO_2$) concentration when airflow is imbalanced: 0.008–0.015 (adjacent cubicle); 0.0015–0.008 (distant cubicle) Normalized tracer gas ($CO_2$) concentration when airflow is balanced: 0.005–0.015 (adjacent cubicle); 0.0015–0.008 (distant cubicle) Normalized tracer gas ($CO_2$) concentration using downward ventilation: 0.003–0.005 (adjacent cubicle); 0–0.003 (distant cubicle) | Airflow balancing associated with a slight improvement in normalized bio-aerosol concentration compared to unbalanced airflow. Downward ventilation associated with a reasonable improvement in normalized bio-aerosol concentration compared to mixing ventilation. *Authors' Recommendations*: "The design of the air-conditioning system in the ward should be improved to minimize cross-infection of airborne respiratory infectious diseases such as SARS and influenza. Testing and commissioning of an air-conditioning system for a hospital ward should be done carefully. Regular checks of the flow balancing was also necessary." (p.94) |
| Lim (Jan 2010) [52] South Korea Modelling; Prince of Wales hospital ward SARS | Supply and exhaust airflow rate vs concentration of tracer gas Diffuser and exhaust location vs concentration of tracer gas | Tracer gas ($N_2O$) concentration in reference case: $45.419 \times 10^{-4}$ ppm (parts per million) (initial bay); $5.861 \times 10^{-4}$ ppm (adjacent bay); $2.027 \times 10^{-4}$ (distant bay 1); $1.054 \times 10^{-4}$ ppm (distant bay 2) Tracer gas ($N_2O$) concentration when airflow is balanced: $12.057 \times 10^{-4}$ ppm (initial bay); $1.911 \times 10^{-4}$ ppm (adjacent bay); $0.795 \times 10^{-4}$ (distant bay 1); $0.675 \times 10^{-4}$ ppm (distant bay 2) Tracer gas ($N_2O$) concentration when air is exhausted at the center of the ward and supplied at the boundary without air balancing: $21.274 \times 10^{-4}$ ppm (initial bay); $2.291 \times 10^{-4}$ ppm (adjacent bay); $1.422 \times 10^{-4}$ (distant bay 1); $0.934 \times 10^{-4}$ ppm (distant bay 2) Tracer gas ($N_2O$) concentration when air is exhausted at the center of the ward and supplied at the boundary with air balancing: $12.057 \times 10^{-4}$ ppm (initial bay); $1.911 \times 10^{-4}$ ppm (adjacent bay); $0.795 \times 10^{-4}$ (distant bay 1); $0.375 \times 10^{-4}$ ppm (distant bay 2) | Airflow balancing associated with decreased dispersion of virus particles. Air supplied from the room boundary and exhausted in the middle of the room associated with greater efficiency of isolating polluted air. |
| Chen (Oct 2011) [53] China Modelling; Prince of Wales hospital ward SARS | Airflow pattern vs bioaerosol distribution (which results in SARS transmission) | Normalized concentrations in the initial cubicle (0.03), adjacent cubicle (0.011) and two distance cubicles (0.004) were the same between the balanced and imbalanced cases. | Airflow balancing associated with little effect on the distribution of bioaerosols in a hospital ward simulated using multi-zone modelling |
| Satheesan (Feb 2020) [54] China Modelling; Hospital ward MERS-CoV | Ventilation rate (air change rate) vs exhausted ratio to corridor and in-ward exposure to pathogens at various exhaust airflow rates $r_e$ = exhausted ratio EA = exhaust air | Exhaust ratio No exhaust Patients 1 and 2: $r_e < 0.05$ at 3 $h^{-1}$ $r_e > 0.25$ at 13 $h^{-1}$ Patient 5 and 6: $0.05 < r_e < 0.1$ at 13 $h^{-1}$ EA = 10% Patient 1 and 2: $r_e = 0.15$ at 13 $h^{-1}$ EA = 50% Patient 1 and 2: $r_e < 0.03$ at 13 $h^{-1}$ Patient 5 and 6: $r_e < 0.006$ exhausted ratio $r_e$ increases with air change rate in the base case. Exposure Risk -patients in beds located at 1.625 m away from the corridor: exposure risk > 0.05 -patients located at 5.875 m away from the corridor: exposure risk < 0.025 | Increased ventilation rate (ACH) associated with significant reduction in infection risk for patients in the ward located farther away from the corridor. Increased ventilation rate (ACH) associated with increase in infection risk of corridor users and its connected amenities. Installation of exhaust grilles in close proximity to each patient associated with significantly reduced individual patient exposure in the ward. Installation of exhaust grilles in close proximity to each patient associated with considerably reduced risk of infection transmission to corridor users and its connected amenities. *Authors' Recommendations*: "it is recommended to provide exhaust grilles in close proximity to a patient, preferably above each patient's bed." (p.8) |

*(Continued)*

**Table 5.** (Continued)

| Author Year; Country | Outcome parameter | Data | Association |
|---|---|---|---|
| Vuorinen (Oct 2020) [55] Finland Modelling; supermarket SARS-CoV-2 | Exhaust ventilation rate and the use of an air mixing device vs concentration of airborne particles | Mixing ventilation device off + 0.18 $m^3s^{-1}$ ceiling exhaust: slowest dilution<br>Mixing ventilation device on + 0.18 or $1.8m^3s^{-1}$ ceiling exhaust: effective dilution which decreased the normalized concentration to $10^{-4}$–$10^{-3}$ further away from the cough<br>Mixing ventilation device off + 1.8 $m^3s^{-1}$ ceiling exhaust: most effective reduction of particle concentration | Increased exhaust ventilation rate and less air mixing associated with greater dilution of airborne particles. |

ACH = air changes per hour; $CO_2$ = carbon dioxide; EA = Exhaust airflow; $N_2O$ = nitrous oxide; SARS = severe acute respiratory syndrome.

within the ward as well as the corridor. Vuorinen et al found that increasing exhaust flow rates and decreasing air mixing was the most effective intervention to reduce infectious particle concentrations [55].

Two studies analyzing both ventilation rate and airflow patterns provided recommendations [51,55]. Recommendations included designing ventilation systems so that cross-infection was minimized, with regular ventilation testing and air balancing checks [51]. Additionally, Satheesan et al recommended exhaust placement near patients, ideally above the head of the patient [54].

## Risk of bias

Twenty of the 23 modelling studies had low risk of bias for all three domains: definition, assumption, validation. Two modelling studies had low risk of bias for assumption and validation but had unclear risk of bias for definition as there was a lack of clarity regarding contribution of fresh air in Augenbraun et al [36], as noted in the differences between Table 2 and Fig 1, and regarding the HEPA filter efficiency in Kennedy et al [45]. All eight experimental studies had low risk of bias for all three domains: selection bias, information bias, confounding (for the comparison of interest to this review).

## Discussion

A review of 32 ventilation and coronavirus studies offered several crucial observations. First, increased ventilation, whether through ventilation rates (ACH, $m^3$/h, $m^3$/min, L/min) or as determined by $CO_2$ levels (ppm), was associated with decreased transmission, transmission probability/risk, infection probability/risk, droplet persistence, and virus concentration, and increased virus removal and efficiency of virus particle removal. Second, increased ventilation rate was associated with decreased risk for longer exposure times. Third, the use of ventilation was associated with better outcomes than no ventilation scenarios. Fourth, airflow patterns were associated with transmission cases. Fifth, HVAC ventilation feature (supply/exhaust or fans) placement was associated with varied particle distribution. As well, changing ventilation rate or using mixing ventilation is not always the only way to mitigate viruses. Finally, while some studies provided recommendations, few provided specific quantification of ventilation parameters suggesting a significant gap in current research.

Increasing the ventilation rate is an obvious solution to decreasing the risk of viral infection [27–30,33,34,37–39,40–42,44,54]. However, there are some caveats. Adhikari et al found that increasing the ventilation rate did not affect the close-range airborne transmission route [29]. This means that an infected person may transmit the virus to close contacts regardless of the

ventilation rate. Increasing the ventilation rate can also lead to wider spread of the virus, sometimes outside of the ventilated space, as suggested by Satheesan et al [54] and Borro et al [33]. This is why the airflow pattern can play a key role in transmission of the virus.

Airflow pattern in a room is governed by the location of diffusers and exhausts and the volume of air supplied and exhausted. The airflow pattern influences the distribution of the airborne virus in the space. Improper ventilation design can help spread viral particles to larger spaces beyond the proximity of the infected individuals. It can also create local hotspots relative to the infected individual [26]. Li et al [51] and Lim et al [52] both found that unbalanced or improperly balanced ventilation could increase the spread of airborne viral particles outside of the index patient's room. However, by manipulating the HVAC system, spread of the virus particles outside of the room can be prevented [48]. Shao et al determined that multiple supply diffusers at a specified ventilation capacity can significantly improve the viral particles removal rate compared to a single diffuser at the same ventilation capacity [26]. Lim et al found that a balanced supply of air at the boundary and exhausting it in the middle of the room prevents the spread of airborne viral particles [52]. The strategic placement and airflow rate of exhaust grilles can be a crucial factor in designing for infection mitigation [54]. Lu et al attributed an outbreak of SARS-CoV-2 in a restaurant located in Guangzhou to the weak exhaust system which led to the continued presence of viral particles in the air [24,25]. In a restaurant located in Jeonju, Korea, Kwon et al found that only visitors in the airflow path were infected with SARS-CoV-2 [47]. Additionally, it only took 6.5 seconds for the viral particles to travel from the infector to the infectee. To combat this, Augenbraun et al suggests situating individuals in different airstreams so each individual is in their own occupied region which minimizes air mixing [36]. You et al [46] and Vuorinen et al [55] agree that mixing ventilation might not be the most suitable from the perspective of infection mitigation and that other patterns such as displacement ventilation should be considered. Maintaining a certain airflow pattern can be challenging. Chen et al found that a temperature difference between two spaces can interrupt the intended airflow pattern and cause unintentional air exchange between two spaces [53]. They suggested that reducing the area of openings by installing curtains would be an effective approach for reducing virus transmission.

From our review of the literature, it is clear that enhancing the ventilation, whether through increasing the ventilation rate or modifying the airflow pattern, is an effective way to reduce airborne viral infection risk. It is recognized that enhancing ventilation across an entire building might not be feasible due to economic or HVAC system limitations. Buonanno et al suggests paying close attention to spaces where the occupants are engaged in high expiratory activities such as singing, speaking loudly, or heavy exercising [32]. If mechanical ventilation is not available or is insufficient, natural ventilation through opening windows and doors should be utilized. Harrichandra et al studied 12 nail salons in New York City and determined that the risk was lowest in the nail salon with the highest ventilation rate and that this nail salon did not have a dedicated exhaust and used natural ventilation [42]. However, natural ventilation would not be feasible in colder weather. Yu et al [28] and Miller et al [41] suggest pairing ventilation with high efficiency filtration and ultraviolet germicidal irradiation where possible. Irrespective of ventilation design, practices such as frequent handwashing [54], surface cleaning [54], and wearing masks [30,37,41,42,45] should not be overlooked.

## Implications for research

Ventilation is an HVAC feature that incorporates and considers many factors such as ventilation rate, airflow patterns, and air balancing. Additionally, ventilation is affected by outside factors such as room size, airflow rates and volume, filtration usage, exhaust and supply ratios,

and number of occupants, to name a few. As such, recommendations with quantified data can be hard to provide. As Li et al noted in their review, insufficient evidence was found to specify and quantify the minimum ventilation requirements in buildings in relation to the airborne transmission of infectious agents [9]. Unfortunately, this remains a gap in the literature. Within this review, only three studies provided specific, quantified recommendations [28,36,49]. These included using an air change rate of 9 h$^{-1}$ for a typical six-bed hospital ward to potentially decrease the inhalation risk of viral particles and maximize energy efficiency [28]; waiting one air change for filtrated environments to six air changes for non-filtrated environments before reoccupying a room previously occupied by a different person [36]; and maintaining a pressure difference between -2 and -25 Pa in negative pressure isolation spaces [49].

While 10 other studies provided recommendations, they were qualitative (rather than quantitative) [24–26,30,40,44,47–48,51,54]. Ventilation rate recommendations include avoiding poorly ventilated spaces [30]; checking to ensure ventilation was operational before allowing occupants to enter a space and using maximum supply rates [44]; increasing ventilation rates and introducing fresh air [40,48]; and decreasing air recirculation [48]. Air pattern recommendations included improving ventilation generally [24,25]; evaluating fan placement in relation to restaurant seating and ensuring frequent ventilation [47]; and optimizing ventilation by increasing ventilation locations [26]. Finally, recommendations that considered ventilation rate and airflow patterns included suggestions to design ventilation systems in a way that minimized cross-infection using regular ventilation testing and air balancing checks [51] and placing exhaust locations near patients, ideally above the head of the patient [54].

Dai et al provided what appears to be quantitative data regarding necessary ACH to bring the infection probability to specific percentages, but they do not refer these recommendations or even suggestions [34]. However, Dai et al showed the potential for more elaborate quantitative recommendations [34]. As such, the current lack of quantitative data shows a research gap within ventilation literature and a potential topic or consideration for future research.

## Implications for practice

In practice, ventilation should be used, i.e., something is better than nothing [30,40,45]. The ventilation requirements of the HVAC system should consider the occupancy of the space [34,39] and the interplay between ventilation rate, exposure time, and infection risk [32,34,42]. While a variety of settings were addressed, a major portion of the studies (15 of 32) discussed hospitals and/or healthcare facilities. As such, it is important to keep in mind that the portion of studies discussing health care settings (in particular hospitals), with their associated relatively high indoor air quality (IAQ), is not representative of typical filtration and ACH in other identified high risk-of-transmission buildings of concern. Considering diminishing return on improving ACH, practical efforts should be directed at the most high-risk sites (low ACH, crowded, and high-risk occupants or activities).

Interestingly, mixing ventilation might not always be the best and other airflow patterns should be considered to lower virus transmission [46,51,55]. These three studies found that alternative airflow patterns were better at negating transmission than mixing ventilation. An important point in most ventilation pattern studies is that they generally require location and characterization of the source, which in practical situations is difficult to ascertain. The decrease in risk available as an outcome is an essential factor to know if innovation and ventilation design are headed in the direction of "smart ventilation" using continuous sensing of occupancy (or even temperature of room occupants) to create control strategies for air supply flow rate and direction. More knowledge of how airflow patterns alone affect risk could lead to systems with real-time feedback control on airflow pattern.

### Strengths and limitations

This systematic review was prepared according to rigorous methods established by the international Cochrane organization [15], and adheres to comprehensive and transparent reporting standards [17]. We followed an *a priori* protocol which is publicly available [13] and we registered the review prior to its conduct [14]. All data collected during the systematic review are reported within the manuscript. We conducted an extensive search to identify all relevant studies; however, due to resource constraints we only included studies reported in English. Nevertheless, we identified and included a large number of relevant studies that were conducted across a wide range of geographic areas (e.g., USA, China, Taiwan, Netherlands, Switzerland, Finland, Italy, Spain, Australia, Romania, South Korea). We were unable to assess for publication bias as we did not conduct meta-analyses due to variation across studies in terms of design, ventilation features examined, outcomes assessed, and results reported. However, we searched the grey literature to identify conference abstracts and recent publications.

### Conclusion

In 2007, Li et al found insufficient evidence to specify and quantify the minimum ventilation requirements in buildings in relation to the airborne transmission of infectious agents, despite finding sufficient evidence to demonstrate an association between transmission of infectious agents and ventilation rate and/or airflow pattern [9]. In the intervening time, the literature on coronaviruses and ventilation reinforces the association between transmission and ventilation rate and/or airflow pattern but reveals the limited progress towards providing quantitative recommendations. The recommendations with quantified data were using an air change rate of 9 $h^{-1}$ for a hospital ward; waiting six air changes or 2.5 hours before allowing different individuals into an unfiltered office with ~2 FCH and one air change for a high-efficiency MERV or HEPA filtered laboratory; and using a pressure difference between -2 and -25 Pa in negative pressure isolation spaces. Qualitative recommendations included using or increasing ventilation, and the introduction of fresh air, using maximum supply rates, avoiding poorly ventilated spaces, assessing fan placement and potentially increasing ventilation locations, and employing ventilation testing and air balancing checks.

### Supporting information

**S1 Checklist.**
(DOC)

### Acknowledgments

We thank Tara Landry and Alison Henry for conducting the peer review of the search strategies. We thank Samuel Ducholke, Kristen Rumbold, Larry Zhong, and Stella Mathews for their involvement in screening studies for inclusion.

### Author Contributions

**Conceptualization:** Gail M. Thornton, Brian A. Fleck, Lisa Hartling.

**Data curation:** Gail M. Thornton, Brian A. Fleck, Emily Kroeker, Dhyey Dandnayak, Natalie Fleck, Lisa Hartling.

**Formal analysis:** Gail M. Thornton, Brian A. Fleck, Emily Kroeker, Dhyey Dandnayak, Natalie Fleck, Lisa Hartling.

**Funding acquisition:** Brian A. Fleck, Lexuan Zhong, Lisa Hartling.

**Investigation:** Gail M. Thornton, Brian A. Fleck, Emily Kroeker, Dhyey Dandnayak, Natalie Fleck, Lisa Hartling.

**Methodology:** Gail M. Thornton, Brian A. Fleck, Lisa Hartling.

**Project administration:** Gail M. Thornton, Brian A. Fleck, Lisa Hartling.

**Supervision:** Gail M. Thornton, Brian A. Fleck, Lisa Hartling.

**Validation:** Gail M. Thornton, Brian A. Fleck.

**Visualization:** Gail M. Thornton, Brian A. Fleck, Emily Kroeker, Dhyey Dandnayak, Natalie Fleck.

**Writing – original draft:** Gail M. Thornton, Brian A. Fleck, Emily Kroeker, Dhyey Dandnayak, Natalie Fleck, Lisa Hartling.

**Writing – review & editing:** Gail M. Thornton, Brian A. Fleck, Emily Kroeker, Dhyey Dandnayak, Natalie Fleck, Lexuan Zhong, Lisa Hartling.

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
