## [Decision Letter · Decision Letter 0]

13 Apr 2022

PGPH-D-21-00860

The impact of heating, ventilation, and air conditioning design features on the transmission of viruses, including the 2019 novel coronavirus: a systematic review of ventilation and coronavirus

Dear Dr. Fleck,

Thank you for submitting your manuscript to PLOS Global Public Health. After careful consideration, we feel that it has merit but does not fully meet PLOS Global Public Health’s publication criteria as it currently stands. Therefore, we invite you to submit a revised version of the manuscript that addresses the points raised during the review process.

We look forward to receiving your revised manuscript.

Kind regards,

Olatunji O Adetokunboh, MD, PhD

Academic Editor

Journal Requirements:

1. Your co-authors, Gail M. Thornton (gthornto@ualberta.ca), Emily Kroeker (ekroeker@ualberta.ca), Natalie Fleck (njfleck@ualberta.ca), and Lisa Hartling (hartling@ualberta.ca), have not confirmed authorship of the manuscript. We have resent them the authorship confirmation email; however please check that the above email address for them is correct and follow up personally to ensure they confirm. Please note that we cannot pass your manuscript to Production until we have received confirmations from all co-authors.

Just in case your co-authors are having difficulty confirming their authorship, you may advise them to send us an email at globalpubhealth@plos.org and we will confirm their authorship on the authors' behalf.

2. Please amend your detailed Financial Disclosure statement. This is published with the article, therefore should be completed in full sentences and contain the exact wording you wish to be published.

State the initials, alongside each funding source, of each author to receive each grant.

3. Please update the completed 'Competing Interests' statement. Please declare all competing interests beginning with the statement “I have read the journal's policy and the authors of this manuscript have the following competing interests:”.

4. Please provide a complete Data Availability Statement in the submission form. If your research concerns only data provided within your submission, please write “All data are in the manuscript and/or supporting information files” as your Data Availability Statement.

5. Please provide separate figure files in .tif or .eps format only and ensure that all files are under our size limit of 20MB.

Additional Editor Comments (if provided):

Dear Authors,

This is a well written systematic review on a very important and critical subject. However, find below few comments:

1. Page 2; Line 20: Write the official name of SARS-CoV-2 first.

2. Page 4; Line 47: Change "[a]irborne".

3. Page 10; Line 166: Write out carbon dioxide in full first.

4. Include the list of abbreviations.

5. State the strengths and limitations of this study.

6. Please state the role of each reviewer.

7. References No 1: The link is inactive.

8. Figure 1: The title should be underneath the flow chart.

Reviewers' comments:

Reviewer's Responses to Questions

**Comments to the Author**

1. Does this manuscript meet PLOS Global Public Health’s publication criteria? Is the manuscript technically sound, and do the data support the conclusions? The manuscript must describe methodologically and ethically rigorous research with conclusions that are appropriately drawn based on the data presented.

Reviewer #1: No

Reviewer #2: Yes

Reviewer #3: Yes

Reviewer #4: Yes

Reviewer #5: Yes

2. Has the statistical analysis been performed appropriately and rigorously?

Reviewer #1: N/A

Reviewer #2: Yes

Reviewer #3: Yes

Reviewer #4: Yes

Reviewer #5: Yes

3. Have the authors made all data underlying the findings in their manuscript fully available (please refer to the Data Availability Statement at the start of the manuscript PDF file)?

Reviewer #1: No

Reviewer #2: No

Reviewer #3: Yes

Reviewer #4: Yes

Reviewer #5: Yes

4. Is the manuscript presented in an intelligible fashion and written in standard English?

Reviewer #1: No

Reviewer #2: Yes

Reviewer #3: Yes

Reviewer #4: Yes

Reviewer #5: Yes

5. Review Comments to the Author

Reviewer #1: -------------------------------------------------------------------------------------------------------------------------------------------------------------------

Reviewer #2: Title: The impact of heating, ventilation, and air conditioning design features on the transmission of viruses, including the 2019 novel coronavirus: a systematic review of ventilation and coronavirus

Comment

Generally, the study is good to be published, however, it needs some correction. The manuscript has not taken on board the title. When reading the manuscript, you don’t find the issues of heating and air conditioning being mentioned.

Suggestion: The title should read as ‘The impact of heating, ventilation, and air conditioning design features on the transmission of viruses, including the 2019 novel coronavirus: a systematic review of ventilation and coronavirus and then the rest of the manuscript should remain as it is OR the manuscript should be rewritten to take on board the issues of heating and air conditioning design features.

Reviewer #3: This manuscript is a very well written and clear systematic review of the effectiveness of ventilation for mitigating transmission of coronavirus. I agree with the conclusions made from the analysis and found no grammatical errors. An enjoyable read of important evidence synthesis. No further comments.

Reviewer #4: The authors have produced a very thorough and well written systematic review of the literature on how HVAC systems influence coronavirus transmission. The analysis and findings are clearly presented and the discussion highlights gaps in the literature which require future research.

Reviewer #5: A useful summary of the currently available information. I have no specific comments or suggestions for improvement or clarity. Looking forward to the paper's publication and the discussion it generates.

6. PLOS authors have the option to publish the peer review history of their article (what does this mean?). If published, this will include your full peer review and any attached files.

**Do you want your identity to be public for this peer review?** For information about this choice, including consent withdrawal, please see our Privacy Policy.

Reviewer #1: No

Reviewer #2: **Yes: **Dr. Joseph C. Hokororo

Reviewer #3: No

Reviewer #4: No

Reviewer #5: **Yes: **Janet M. Macher

---

## [Editor Report · Decision Letter 1]

9 May 2022

The impact of heating, ventilation, and air conditioning design features on the transmission of viruses, including the 2019 novel coronavirus: a systematic review of ventilation and coronavirus

PGPH-D-21-00860R1

Dear Dr Fleck,

We are pleased to inform you that your manuscript 'The impact of heating, ventilation, and air conditioning design features on the transmission of viruses, including the 2019 novel coronavirus: a systematic review of ventilation and coronavirus' has been provisionally accepted for publication in PLOS Global Public Health.

Best regards,

Olatunji O Adetokunboh, MD, PhD

Academic Editor
